# Increased blood meal size and feeding frequency compromise *Aedes aegypti* midgut integrity and enhance dengue virus dissemination

**Rebecca M. Johnson, Duncan W. Cozens, Zannatul Ferdous, Philip M. Armstrong, Doug E. Brackney**  *

Connecticut Agricultural Experiment Station, Department of Entomology, Center for Vector Biology and Zoonotic Diseases, New Haven, Connecticut, United States of America

* Doug.Brackney@ct.gov

**Data Availability Statement:** All data are in the manuscript and/or supporting information files.

## Abstract

*Aedes aegypti* is a highly efficient vector for numerous pathogenic arboviruses including dengue virus (DENV), Zika virus, and yellow fever virus. This efficiency can in part be attributed to their frequent feeding behavior. We previously found that acquisition of a second, full, non-infectious blood meal could accelerate virus dissemination within the mosquito by temporarily compromising midgut basal lamina integrity; however, in the wild, mosquitoes are often interrupted during feeding and only acquire partial or minimal blood meals. To explore the impact of this feeding behavior further, we examined the effects of partial blood feeding on DENV dissemination rates and midgut basal lamina damage in *Ae. aegypti*. DENV-infected mosquitoes given a secondary partial blood meal had intermediate rates of dissemination and midgut basal lamina damage compared to single-fed and fully double-fed counterparts. Subsequently, we evaluated if basal lamina damage accumulated across feeding episodes. Interestingly, within 24 hours of feeding, damage was proportional to the number of blood meals imbibed; however, this additive effect returned to baseline levels by 96 hours. These data reveal that midgut basal lamina damage and rates of dissemination are proportional to feeding frequency and size, and further demonstrate the impact that mosquito feeding behavior has on vector competence and arbovirus epidemiology. This work has strong implications for our understanding of virus transmission in the field and will be useful when designing laboratory experiments and creating more accurate models of virus spread and maintenance.

## Author summary

*Aedes aegypti* is an important vector of many arboviruses that profoundly affect human health. Despite its importance as a vector, the impact of field-realistic *Ae. aegypti* feeding behavior on virus transmission is poorly understood. In this study, we investigated the physiological impacts of blood meal size and number on vector competence. We found

**Funding:** This work was supported in part by grants from the National Institutes of Health, National Institute of Allergy and Infectious Diseases (AI148477) (DEB and PMA). The funders had no role in study design, data collection and analysis, decision to publish, or preparation of the manuscript.

**Competing interests:** The authors do not have any competing interests that could be perceived to bias this work.

that partial blood meals—common when mosquitoes feed on defensive hosts—are sufficient to increase mosquito midgut basal lamina damage and dengue virus dissemination. Nevertheless, virus dissemination still occurred in the absence of heightened midgut damage, suggesting that mosquito midguts are not impermeable to virus escape. We also observed that frequent successive blood feeding results in slightly higher levels of midgut damage with each instance of feeding. These experiments more closely mimic mosquito feeding behavior in field settings and suggest that current studies often underestimate the impact of mosquito feeding behavior on virus dissemination and transmission dynamics.

## Introduction

*Aedes aegypti* mosquitoes have a wide geographic distribution and are highly efficient vectors for many pathogenic arboviruses including dengue virus (DENV), Zika virus (ZIKV), and yellow fever virus (YFV). Some of this efficiency is due to their high feeding frequency as *Ae. aegypti* mosquitoes bite humans an estimated 0.63–0.76 times per day and sometimes take partial rather than full blood meals [1]. Despite this, laboratory experiments assessing vector competence and extrinsic incubation period (EIP)—the length of time between ingestion of virus to when a mosquito can transmit virus via an infectious bite—typically provide a single blood meal, allow mosquitoes to feed to repletion, and then temporally sample bodies, legs, and saliva to measure infection, dissemination, and transmission status, respectively. Previously, we found that feeding frequency alters virus transmission dynamics and that feeding mosquitoes a second, full, non-infectious blood meal three days after an initial infectious feed, led to faster virus dissemination and a shorter EIP [2]. This shorter EIP corresponds to a transient increase in midgut basal lamina disorder and damage following blood feeding that may explain the speed at which virus particles can traverse the midgut escape barrier when mosquitoes are fed a second time [2–5]. Although these previous studies have been important for estimating vectorial capacity and indicate that frequency of feeding is important, they do not accurately reflect mosquito feeding behavior in the wild where mosquitoes often encounter defensive hosts and take partial or multiple interrupted blood meals. The interplay between blood meal size, basal lamina damage, and midgut escape or virus dissemination has been underexplored.

  Feeding attempts on defensive hosts can lead to feeding disruption and acquisition of partial or minimal blood meals. Mosquitoes that are initially unsuccessful in reaching repletion may try to feed again within the same gonotrophic cycle, leading to multiple or mixed blood meals, higher host biting rates, and more opportunities for mosquitoes to acquire or transmit viruses [6]. Stretch receptors and the degree of midgut distention determine whether mosquitoes continue to host-seek following disruption and factors such as host availability and defenses determine refeeding success [7]. In laboratory experiments, *Ae. aegypti* provided blood meals less than 2.5 μl continued to host-seek whereas larger blood meals blocked this behavior [8]. Field collections have reported high levels of refeeding or multiple feeding within the same gonotrophic cycles. In Thailand, 0–32% of engorged *Ae. aegypti* contained double human blood meals, depending on the year, season, and village [9]. Another study reported that 32% of engorged mosquitoes in Puerto Rico and 42% in Thailand had taken double blood meals while an additional 2% and 5%, respectively, had taken triple blood meals [1]. While rates of true partial feeding in field conditions are understudied and degree of engorgement is often not scored, partial feeding rates are likely high due to host defensive behaviors and the well documented high levels of multiple blood meals. Therefore, it is important to study the impact of partial blood meals on basal lamina integrity and subsequent virus dissemination.

The mosquito midgut escape barrier is largely composed of the basal lamina that surrounds the midgut and is primarily made up of type IV collagen and laminin with lesser amounts of nidogen/entactin and proteoglycans of the perlecan type [10,11]. These proteins assemble into a grid-like matrix visible using TEM with laminin forming the primary layer that is attached to the midgut epithelial cells via integrin and collagen forming polymerized sheet-like layers that are anchored to laminin and each other with perlecan and nidogen [10,11]. The resulting basal lamina layer has a pore size of ~9–11 nm, yet viruses with larger diameters such as DENV (50 nm) still cross this barrier to disseminate to peripheral tissues within the mosquito [4,12,13].

The mechanism of midgut virus escape remains unclear but a variety of possible routes have been proposed including through the tracheal system, visceral muscles, or by traversing the basal lamina layer due to enzymatic activity or mechanical damage during blood feeding [5,12,14,15]. Mosquito blood feeding induces swift midgut expansion and transient basal lamina collagen IV damage which supports virus escape via mechanical damage of the basal lamina layer [2]. Additionally, virus particles have been observed inside basal lamina layers following blood feeding using TEM [3,5]. Based on these data, our working understanding of virus dissemination is as follows. In mosquitoes given a single blood meal, basal lamina damage heals quickly and is decreasing as midgut epithelial cell infection is establishing and viral replication is beginning, thus lowering the chance of virus particles quickly traversing the basal lamina. If fed a second time a few days later, basal lamina damage again transiently increases but now virus infection is established in proximity to the basal lamina layer, thus increasing the probability of virus particles encountering areas of damage [4]. Dissemination prevalence in mosquitoes given a single blood meal eventually catches up to those in double fed mosquitoes, indicating that the midgut basal lamina layer may not fully repair following blood feeding or that it may have a baseline level of damage or turnover that replicating virus particles eventually encounter. Mosquitoes given a noninfectious blood meal before an infectious feed do not have a reduced EIP which suggests that midgut basal lamina repair following blood feeding is able to return the midgut to a pre-feeding state and that the timing of repair and chance of virus escape are connected [2].

Our earlier experiments clearly demonstrate that secondary blood meals hasten virus midgut escape and dissemination within the mosquito [2]. Nevertheless, in these assays, mosquitoes were given full blood meals whereas mosquitoes in the wild are frequently disrupted during feeding attempts and may not successfully reach repletion [1,6,9]. It is unclear how partial or minimal blood meals may influence basal lamina integrity and arbovirus dissemination or if other routes play a role in virus midgut escape. In this study we examined basal lamina damage under unfed, minimally fed, partially fed, and fully fed conditions and examined the temporal aspects of basal lamina repair following single or multiple rounds of blood feeding. We further investigated the connection between basal lamina damage and midgut escape by looking at dengue serotype 2 (DENV-2) dissemination in mosquitoes given partial or full secondary blood meals as well as in minimally fed mosquitoes to mimic a naïve or unfed state.

## Methods

### Mosquito rearing and feeding

*Ae. aegypti* (Orlando strain, collected from Orlando, FL in 1952) mosquitoes were reared at 27°C with a 14:10 light: dark cycle. Larvae were separated into trays of ~200 per liter of water and were fed with a (3:2) mix of powdered liver: yeast. Cages of adults were fed 10% sugar and starved overnight before being offered blood. In all experiments, female mosquitoes that were approximately one week old were fed on defibrinated sheep's blood using glass water-jacketed membrane feeders. Fully fed mosquitoes were allowed to feed to repletion, partially fed

mosquitoes were aspirated after midgut expansion was observed, minimally fed mosquitoes were fed for 15 seconds (counted after first probing), and control mosquitoes were unfed. Mosquitoes were sorted on ice and placed into cups.

## Collagen hybridization peptide assays

To compare midgut basal lamina damage between treatments, mosquito midguts were pooled into groups of 3 or 4 per tube, depending on experiment. Midguts were dissected at 0 hours post blood meal (hpbm), 24 hpbm, 96 hpbm, or concurrently if unfed. Time points were chosen based on previous data showing that shortly following blood feeding, damage was high (0 hpbm, 24 hpbm), but by 96 hpbm, digestion had completed and damage had returned to a baseline, pre-feeding level [2]. Briefly, dissected midguts were fixed for 24 hours in 2.5% glutaraldehyde and 2% paraformaldehyde, washed 3X with PBS, blocked for 1 hour with 5% Bovine Serum Albumin, washed 1X with PBS, and then incubated overnight at 4°C with 5 μM 5-FAM conjugated collagen hybridizing peptide (CHP) (3Helix; FLU300) as described previously [2]. Previous work has shown this probe to bind specifically to denatured collagen including that found in the mosquito midgut where the major form of collagen is collagen IV [16–18]. As such, this probe was used to assess midgut basal lamina collagen IV denaturation and damage following blood feeding. Following incubation, samples were washed 5X with PBS to remove excess unbound CHP and then incubated in 1μg/μl elastase in PBS for 2 hr at 27° C while agitating every 15 minutes to dislodge bound CHP. Supernatant containing CHP was transferred to a 96-well plate, diluted 1:1 with PBS, and fluorescence was measured using a BioTek SYNERGY H1 microplate reader using the area scan feature and an excitation/emission of 485/515 nm. Collagen IV damage measurements were pooled from three independent replicates for each experiment except the minimal blood feed assay where one replicate was conducted.

## TEM imaging and analysis

To prepare samples for TEM, midguts were dissected in 0.1M cacodylate buffer with 4% paraformaldehyde (PFA) and fixed in 2.5% glutaraldehyde, 2% PFA for 2 hours at room temperature before being placed at 4°C. Samples were held at 4°C for 1–2 days, rinsed with 0.1M cacodylate buffer and then processed and embedded in resin by the Yale Center for Cellular and Molecular Imaging (CCMI) Electron Microscopy Facility. Sections were sliced and placed on copper grids by the CCMI. Images were acquired on a FEI Tecnai BioTwin or Hitachi HT7800 scope. For each treatment group, guts from two individual mosquitoes were sectioned and imaged. As basal lamina thickness may vary between mosquitoes or area of the gut observed, 5 TEM images were chosen for each mosquito (for a total of 10 images per treatment) and the basal lamina width was measured using the straight-line tool and measure function in ImageJ. Basal lamina width measurements—distance between the midgut cell boundary and the outermost basal lamina sheet—were taken in 5 randomly selected locations per image to assess disorder. These measurements were averaged for each image to give 10 measurements per treatment.

## Virus and cell culture

*Aedes albopictus* C6/36 cells were maintained through passaging at a 1:15 dilution in T75 flasks using minimum essential medium containing 2% fetal bovine serum (FBS), 1X non-essential amino acids, 100 U/ml penicillin, and 100 μg/ml streptomycin. For virus infection and production, cells were grown to 70–80% confluency in T75 flasks and infected using 250 μl DENV-2 virus stocks (125270/VENE93; GenBank: U91870) diluted in 3 ml of media. Flasks

were incubated for 1 hr on a rocking platform before adding 12 ml media. Infected cells were held at 28˚C with 5% $CO_2$ for 5 days. Cellular supernatant containing DENV-2 was removed from flasks and diluted 1:5 in defibrinated sheep's blood before feeding to mosquitoes for minimal feeding experiments and experiments looking at the impact of second blood meal size on dissemination. Titers of live virus fed to mosquitoes were quantified as focus forming units per milliliter (FFU/mL) using focus forming assay as described below. Titers ranged from $4.15 \times 10^6$–$1.65 \times 10^7$ FFU/mL.

## Focus forming assays

*Aedes albopictus* C6/36 cells were seeded into 96-well plates at a density of $3 \times 10^5$ cells/well and incubated overnight at 28˚C with 5% $CO_2$. Virus aliquots used to feed mosquitoes were serially diluted ten-fold in serum-free minimum essential medium and 30 μl of each dilution was used to infect wells for 1 hour at 28˚C with 5% $CO_2$. Infectious media was removed and cells were covered with 100 μl of 1% methylcellulose in maintenance media and incubated for 3 days. Cells were then fixed for 15 minutes at room temperature with 100 μl of 4% formaldehyde in PBS, washed 3 times with 100 μl PBS, and permeabilized with 0.2% Triton-X in PBS for 10 minutes at room temperature. Cells were again washed and 30 μl of mouse anti-flavivirus group antigen antibody (D1-4G2-4-15 (4G2)) (NovusBio NBP2-52709) diluted 1:500 in PBS was added to each well. After an overnight incubation at 4˚C, cells were washed and 30 μl of goat anti-mouse IgG (H+L) cross-adsorbed secondary antibody, Alexa Fluor 488 (Invitrogen A11001) diluted 1:200 in PBS was added to each well and cells were incubated overnight at 4˚C. Plates were washed to remove excess secondary antibody and foci were counted using a Zeiss Axio Vert.A1 inverted microscope with a 2.5X objective and a FITC filter.

## Infection and dissemination assays

One week old adult female mosquitoes were starved overnight before being fed with DENV-2 diluted 1:5 in defibrinated sheep's blood. For assays looking at the impact of second blood meal size on dissemination, mosquitoes were given a full infectious blood meal and were sorted on ice. Mosquitoes were returned to cages with an egg laying cup and given either no additional blood meal or an additional non-infectious full or partial blood meal 3 days later. Bodies and legs were harvested at 7 days post infection (dpi) to assess infection and dissemination frequencies. For assays looking at infection and dissemination with minimally fed mosquitoes, mosquitoes were fed either a full blood meal containing DENV2 and sorted on ice or individually fed a minimal infectious blood meal for 15 seconds (counted after first probing). Bodies and legs were harvested 10 days post infection (dpi) to assess infection and dissemination frequencies. For both assays, legs and bodies were homogenized and extracted using a Mag-Bind Viral DNA/RNA 96 Kit (Omega Bio-tek Inc., Norcross, GA) and a Kingfisher Flex automated nucleic acid extraction device (ThermoFisher Scientific, Waltham, MA) as per manufacturer instructions. DENV-2 infection and dissemination frequencies were measured using an iTaq Universal Probes One-Step Kit (BioRad, 1725141) and 20 μl reactions with DENV-2 Fwd: CATGGCCCTKGTGGCG and DENV-2 Rev: CCCCATCTYTTCAG-TATCCCTG primers and DENV-2 Probe: [FAM] TCCTTCGTTTCCTAACAATCC [BHQ1]. Conditions for qPCR reactions were 50C for 30 min, 95C for 10 min and 40 rounds of 50C for 15 sec and 60C for 1 min. A cut-off value used for samples to be considered positive by RT-qPCR was Ct<36. For minimal blood feeding assays two biological replicates were conducted. Three biological replicates were conducted for assays looking at the impact of second blood meal size on dissemination.

## Statistical analyses

For CHP assays, normality was determined using a Shapiro-Wilk test. In all cases except the minimally fed versus fully fed assay and the assay examining the impact of blood meal number on collagen IV damage at 24 hpbm, differences were evaluated using a one-way ANOVA with a Tukey's multiple comparisons test comparing the mean of each group. For comparing collagen IV damage between minimally and fully fed mosquitoes, groups were compared using an unpaired t-test. For comparing differences in collagen IV damage 24 hpbm in mosquitoes fed 1, 2, or 3 blood meals, differences were assessed using a Kruskal-Wallis test with a Dunn's multiple comparisons test comparing mean ranks between groups. Differences in basal lamina thickness measurements were assessed using a Shapiro-Wilk test followed by a Kruskal-Wallis test with a Dunn's multiple comparisons test comparing mean ranks between groups. For DENV-2 infection and dissemination assays, differences between groups were quantified using Fisher's exact tests and the standard error of sample proportions was used to construct standard error bars. All analyses were performed using GraphPad Prism Statistical software and additional descriptive statistics are provided in text and figure legends.

## Results

Laboratory assessments of mosquito digestion and vector competence often use mosquitoes that have fed to repletion; however, this does not accurately capture the range of engorgement levels observed in field settings. To quantify the impact that partial feeding has on basal lamina integrity, *Ae. aegypti* midguts were examined using a CHP binding assay and TEM following acquisition of either no blood meal, a partial blood meal, or a full blood meal. Immediately following blood feeding, mosquito midguts swell dramatically with guts from partially fed mosquitoes exhibiting less expansion than those from mosquitoes given a full blood meal (Fig 1A). As previously observed, mosquitoes that were fully fed (Full) had significantly higher levels of midgut basal lamina collagen IV damage 24 hpbm compared to the unfed (Unfed) controls (Fig 1B; Full vs Unfed p<0.0001). Similarly, mosquitoes that were partially fed (Partial) displayed intermediate levels of damage that were significantly different from both the unfed and fully engorged (Fig 1B; Unfed vs Partial p<0.0001; Full vs Partial p<0.0001). To confirm these results, we qualitatively analyzed midguts using TEM imaging 24 hpbm and measured basal lamina width as a proxy for damage. Visualization of the unfed gut reveals a tightly-packed sheet-like network of basal lamina (Fig 1C). Proportional to the size of the blood meal, the sheet-like network of the basal lamina becomes increasingly disordered and less dense in partial and fully fed groups (Fig 1C). Using ImageJ, we quantified the thickness of the basal lamina sheet-network for each group to measure damage and disorder. As with the CHP assay, the fully fed group had a significantly wider basal lamina and higher degree of disorder compared to the unfed group (Fig 1D; Unfed vs Full p = 0.0017). An intermediate degree of disorder and width was observed in the partially fed group that was not significantly different from the fully fed and unfed groups (Fig 1D; Unfed vs Partial p = 0.4647; Full vs Partial p = 0.1265).

Previously we found that giving mosquitoes a second blood meal three days after an initial full infectious feed led to earlier virus dissemination and a shorter EIP despite having no impact of DENV titer [2]. To assess the effect of varying sizes of secondary blood meal, we measured midgut collagen IV damage 24 hours after a secondary blood meal or equivalent time point if not given a secondary blood meal and quantified DENV-2 infection and dissemination 7 dpi (Fig 2A). In concordance with data from mosquitoes fed a single blood meal, midgut collagen IV damage was proportional to secondary blood meal size and mosquitoes that were given an initial single full blood meal (Single-full) without a secondary blood meal had significantly less midgut basal lamina collagen IV damage than those given two full blood

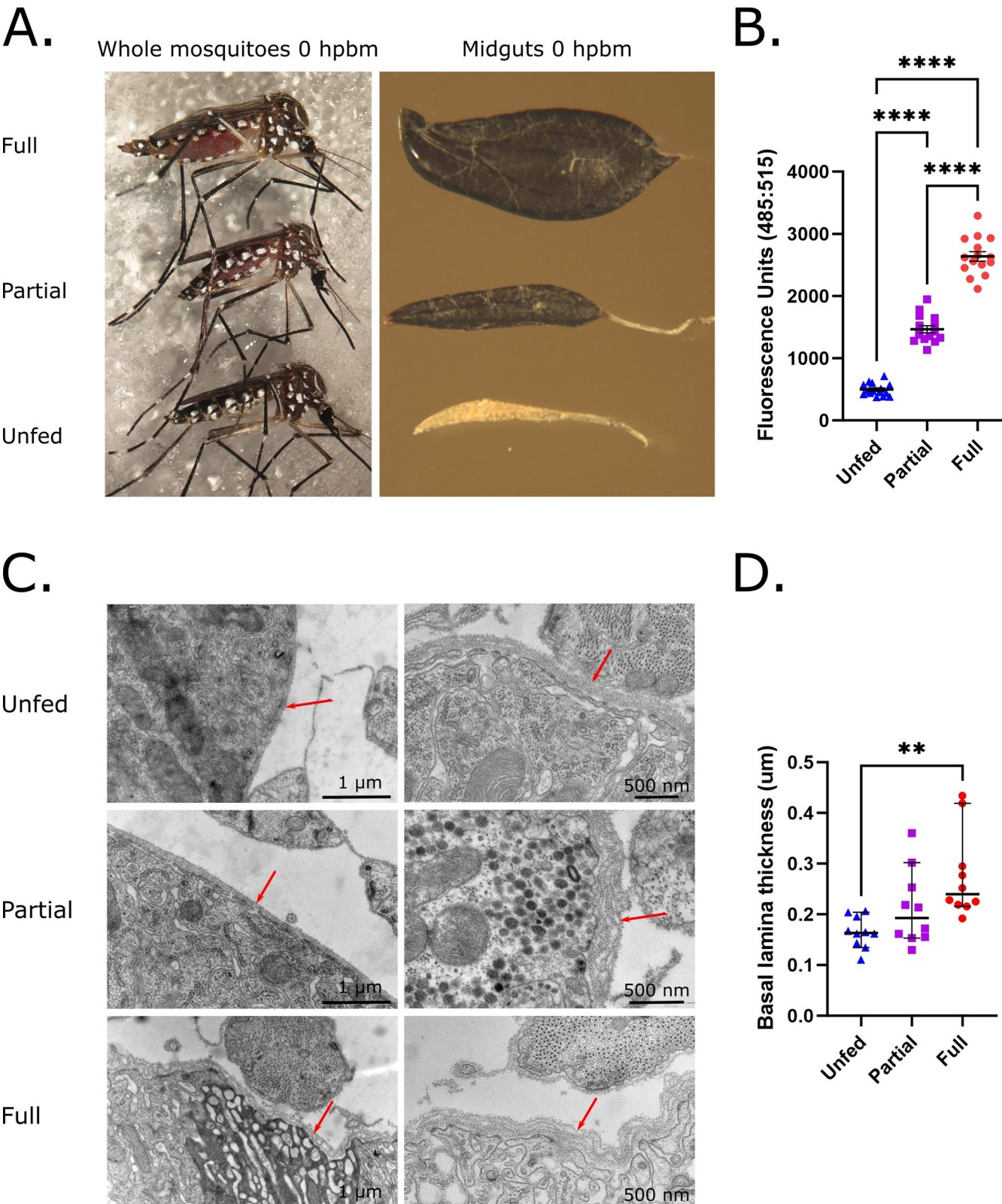

**Fig 1. Partial blood meals result in intermediate levels of midgut expansion, collagen IV damage, and basal lamina disorder.** (A) Whole mosquitoes and dissected midguts 0 hpbm when given a full blood meal (Full), partial blood meal (Partial), or no blood meal (Unfed). (B) Midgut collagen IV damage in unfed,

partially fed, and fully fed mosquitoes 24 hpbm as assessed through CHP binding to pools of 3 midguts (n = 15 pools/treatment of 3 midguts each). Differences were measured using a one-way ANOVA with a Tukey's multiple comparisons post-test. Center lines represent means and bars represent SEM. (C) TEM images of midgut basal lamina (arrows) sections from unfed, partially fed, and fully fed mosquitoes 24 hpbm. Scale bars are 1μm and 500 nm. (D) Basal lamina thickness measurements from two mosquitoes per treatment with 5 TEM images per mosquito and 5 measurements per image that were averaged for 10 measurements per treatment. Differences between treatments were evaluated using a Kruskal-Wallis test with a Dunn's multiple comparisons test comparing mean ranks between groups. Center lines represent median values and error bars represent a 95% confidence interval. On graphs, n.s. = not significant, (*) $p<0.05$, (**) $p<0.01$, (***) $p<0.001$, and (****) $p<0.0001$. Exact p values can be found in text.

meals (Double-full) (Fig 2B Single-full vs Double-full $p<0.0001$). When mosquitoes were given an initial full blood meal followed by a secondary partial blood meal three days later (Full-partial), midgut collagen IV damage 24 hpbm was significantly greater than in mosquitoes that were not given a secondary blood meal but less than in mosquitoes given a full secondary blood meal (Fig 2B; Full-partial vs Single-full $p<0.0001$; Full-partial vs Double-full $p<0.0001$). To examine the impact of secondary blood meal size and varying basal lamina damage on virus dissemination, mosquitoes that were infected with DENV-2 through an initial full blood meal were fed a second time three days later and, as before, were given either a full (Double-full), partial (Full-partial), or no secondary blood meal (Single-full). In agreement with previous data, infection prevalence 7 dpi was the same regardless of secondary blood meal presence or volume (Fig 2C; Single-full vs Double-full p = 0.1299; Single-full vs Full-partial p = 0.1735; Full-partial vs Double-full $p>0.9999$). The acquisition of a secondary blood meal of any size had a significant impact on the prevalence of dissemination observed at 7 dpi. Mosquitoes receiving no secondary blood meal had a lower prevalence of dissemination than those that received either a partial or full secondary blood meal (Fig 2D; Single-full vs Full-partial p = 0.0176; Single-full vs Double-full $p<0.0001$), indicating that secondary partial blood meals were sufficient to enhance dissemination. Mosquitoes that received a partial or full secondary blood meal did not significantly differ from each other in the percentage of mosquitoes with disseminated infection at 7 dpi (Fig 2D; Full-partial vs Double-full p = 0.0911) but the overall trend indicates that dissemination may be proportional to secondary blood meal volume.

Our working model posits that breaks in the basal lamina facilitate virus escape from the midgut. However, mosquitoes provided a single blood meal can still eventually develop disseminated infections despite the basal lamina returning to pre-feed levels of damage as determined by our CHP assay. This paradox raises the question: Do unfed mosquito midguts have baseline levels of basal lamina damage or are the absorbance values observed in unfed midguts the result of non-specific CHP binding? To address this, we provided mosquitoes a minimal blood meal to simulate a near naïve midgut state. Comparison of midguts between unfed and minimally fed mosquitoes revealed no qualitative difference in midgut distention at 0 hpbm (Fig 3A). Quantitative comparisons using our CHP assay also found no significant difference between those provided a minimal blood meal (Minimal) and the unfed controls (Unfed) (Fig 3B; Minimal vs Unfed p = 0.6612). These data demonstrate that providing mosquitoes with a minimal blood meal does not significantly change basal lamina associated collagen IV damage. Thus, this approach can be utilized to introduce virus into mosquitoes and examine virus escape without elevated blood meal associated damage. DENV-2 diluted 1:5 in defibrinated sheep's blood was fed to *Ae. aegypti* mosquitoes via a minimal blood meal and, as a control, a second cohort was allowed to feed to repletion on a normal dose of virus. Bodies and legs were collected from both groups 10 days post blood meal and analyzed for the presence of DENV-2 by qRT-PCR. Despite imbibing very little blood, 19% (5/26) of those receiving a minimal blood meal became infected (Fig 3C) whereas 70% (19/27) of the fully fed controls became infected with a 63% (12/19) dissemination rate (Fig 3D). Interestingly, 2 of 5 infected

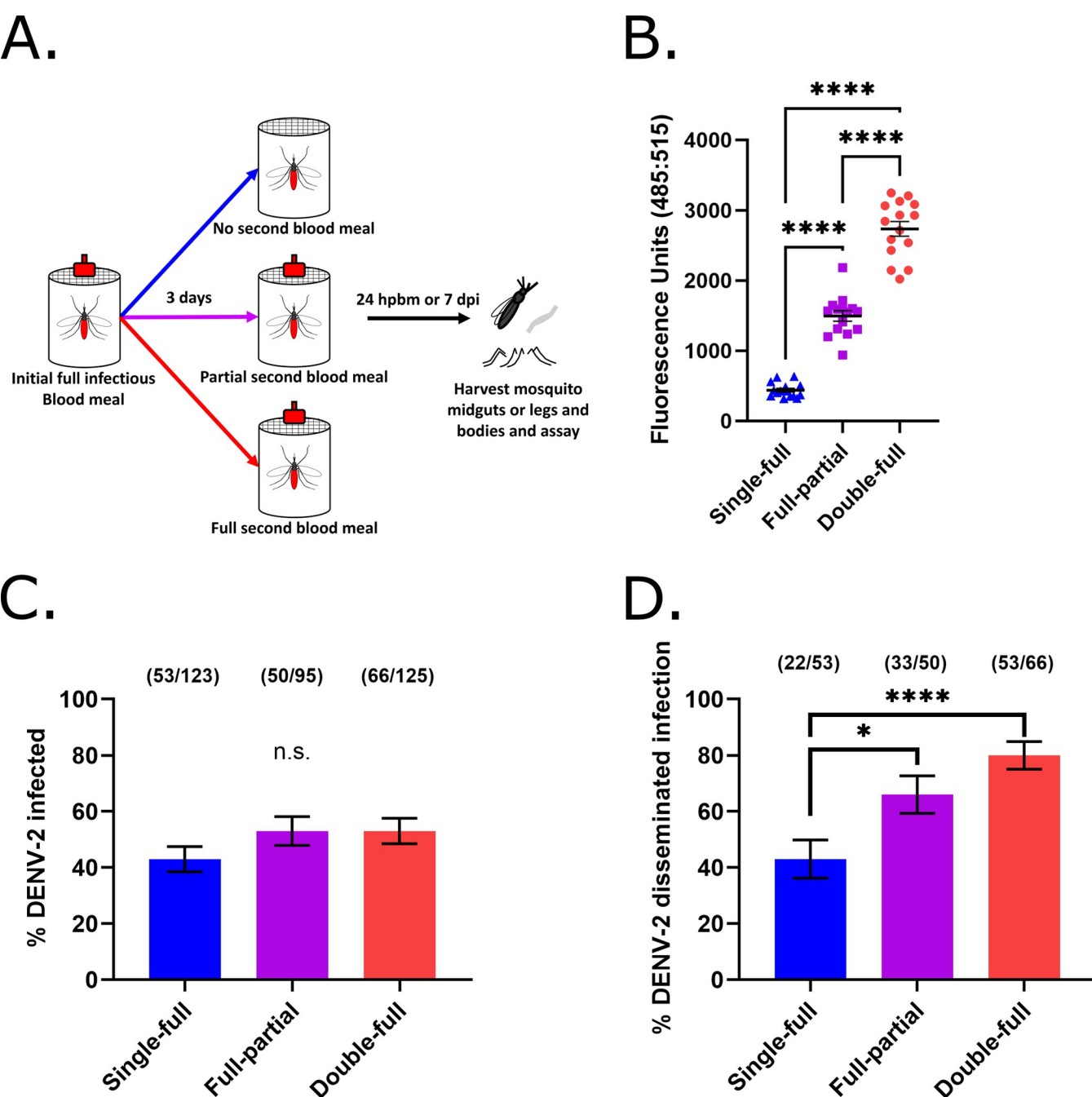

**Fig 2. Secondary partial blood meals are sufficient to induce midgut basal lamina damage and a higher prevalence of DENV-2 dissemination.** (A) Experimental design schematic for double fed mosquitoes. (B) Midgut collagen IV damage 24 hours after the time of the second blood meal in mosquitoes given a single full blood meal (Single-full), a full blood meal and then a partial blood meal (Full-partial), or two full blood meals (Double-full). CHP binding was assessed using pools of 3 midguts (n = 15 pools/treatment of 3 midguts each) and differences were measured using a one-way ANOVA with a Tukey's multiple comparisons post-test. Center lines represent means and bars represent SEM. (C) Percent of mosquitoes infected with DENV-2 at 7 dpi as assessed through testing mosquito bodies. Differences were evaluated using a two-tailed Fisher's exact test. (D) Percent of mosquitoes with disseminated DENV-2 infection 7 dpi as assessed through testing mosquito legs. Data were analyzed using a two-tailed Fisher's exact test. For C and D, center values represent the proportion and bars represent the binomial SE of sample proportions. On graphs, n.s. = not significant, (*) p<0.05, (**) p<0.01, (***) p<0.001, and (****) p<0.0001. Exact p values can be found in text.

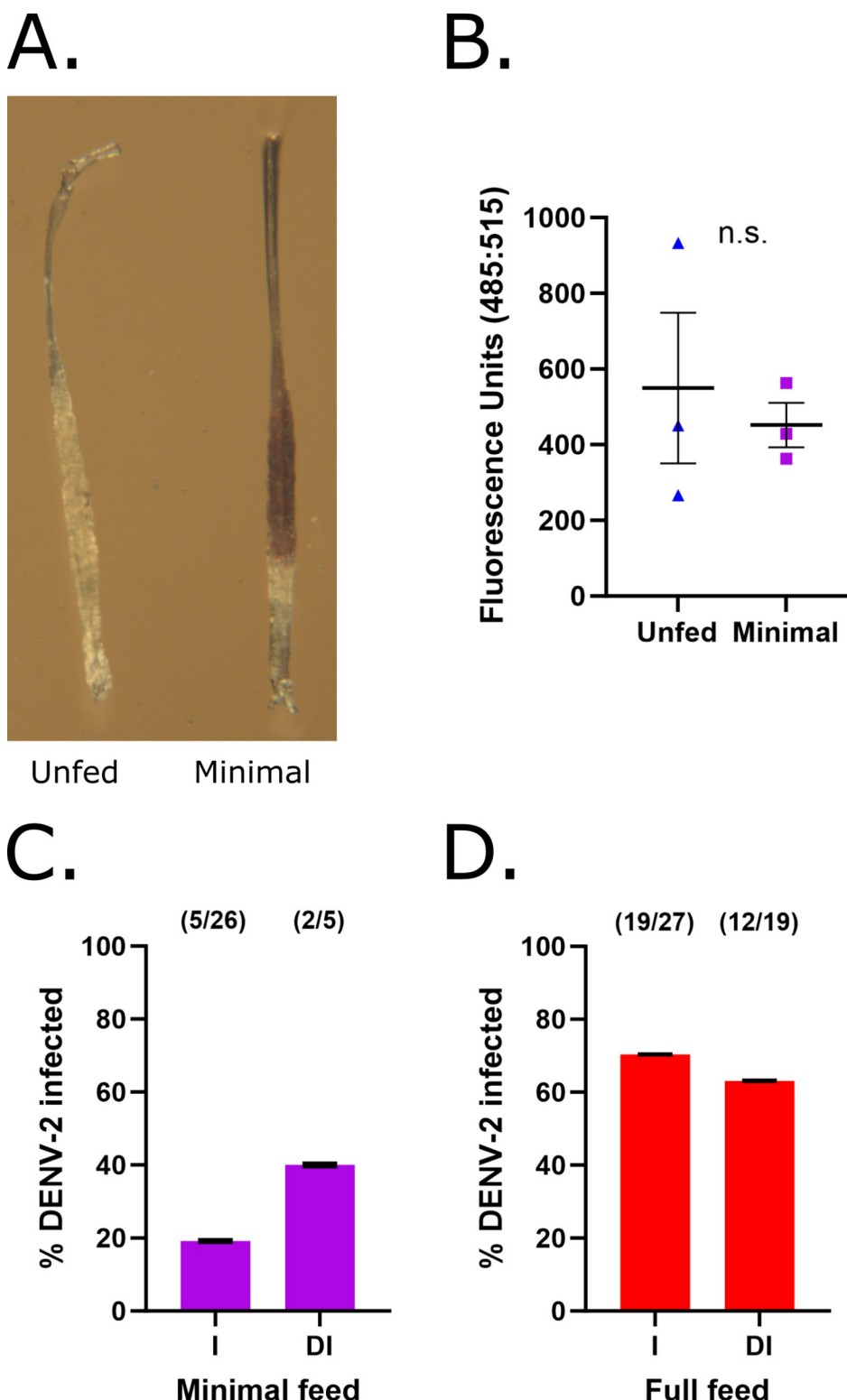

**Fig 3. Midgut basal lamina damage is comparable in naïve and minimally fed midguts, yet minimal blood meals still lead to infection and dissemination.** (A) Mosquito midguts 0 hpbm when given either no blood meal (Unfed) or a minimal 15 sec blood meal (Minimal). (B) Midgut basal lamina collagen IV damage 0 hpbm. CHP binding was assessed using 3 pools/treatment of 4 midguts each (n = 3) and an unpaired t-test. Center lines represent means and bars represent SEM. (C) Percent of minimally fed mosquitoes with a DENV-2 infection (I) or disseminated infection

(DI) 10 dpi as assessed through testing mosquito bodies and legs, respectively. (D) Percent of fully fed mosquitoes with a DENV-2 infection (I) or disseminated infection (DI) 10 dpi. For C and D, center values represent the proportion positive for DENV-2 and bars represent the binomial SE of sample proportions. On graphs, n.s. = not significant, (*) p<0.05, (**) p<0.01, (***) p<0.001, and (****) p<0.0001. Exact p values can be found in text.

mosquitoes from the minimally fed group also developed disseminated infections (Fig 3C). Together, these data suggest that midgut basal lamina integrity is not absolute and that these imperfections can be exploited during the course of infection with or without additional transient basal lamina damage due to blood feeding.

To further investigate basal lamina repair after blood feeding as well as to study the impact of *Ae. aegypti* feeding frequency, we examined collagen IV damage in unfed mosquitoes as well as in mosquitoes 24 and 96 hpbm after 1, 2, or 3 blood meals (Fig 4A). At 24 hpbm, collagen IV damage in mosquitoes given, 1 (BM1), 2 (BM2), or 3 (BM3) blood meals was higher than in mosquitoes that were unfed (UF) and trended upwards with increasing blood meal number (Fig 4B; UF vs BM1 p = 0.0019, UF vs BM2 p<0.0002, UF vs BM3 p <0.0001). No significant differences were detected when comparing mosquitoes given 1 blood meal versus 2 blood meals (Fig 4B; BM1 vs BM2 p >0.9999), 1 blood meal versus 3 blood meals (Fig 4B; BM1 vs BM3 p = 0.9099), or 2 blood meals versus 3 blood meals (Fig 4B; BM2 vs BM3 p >0.9999). Despite this lack of significance between treatments, damage did increase slightly with increasing blood meal number and the variation in damage (as measured using standard deviation) also increased (Fig 4B; UF SD = 112.2, BM1 SD = 153.1, BM2 SD = 265.3, BM3 SD = 364.1). Despite this link between blood meal number and midgut damage at 24 hpbm, by 96 hpbm all groups had recovered and had equivalent levels of collagen IV damage as detected via CHP assay (Fig 4C; UF vs BM1 p = 0.9895, UF vs BM2 p = 0.9922, UF vs BM3 p = 0.8568, BM1 vs BM2 p = 0.9395, BM1 vs BM3 p = 0.9694, BM2 vs BM3 p = 0.7166).

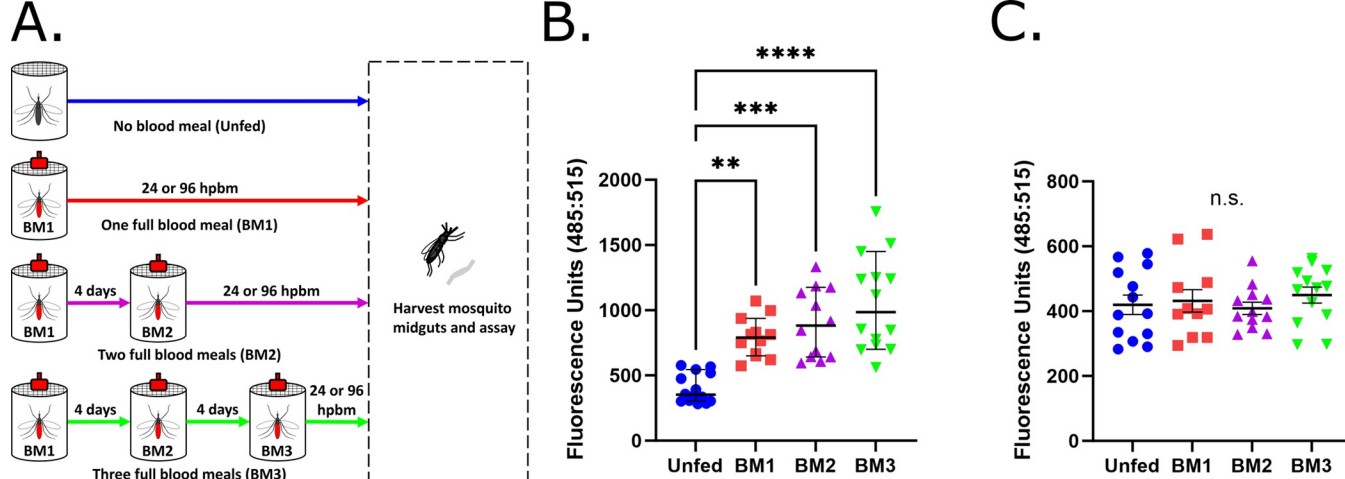

**Fig 4. Basal lamina damage 24 hpbm increases with blood meal number but repairs by 96 hpbm.** (A) Experimental design schematic for mosquitoes given multiple successive blood meals. (B) Midgut collagen IV damage 24 hours after 1 (BM1), 2 (BM2), 3 (BM3), or no blood meal (Unfed). CHP binding assays used pools of 3 midguts with 12–14 pools/treatment (Unfed n = 14, BM1 n = 12, BM2 n = 12, BM3 n = 14). Data was analyzed using a Kruskal-Wallis test with a Dunn's multiple comparisons test comparing mean ranks between groups. Center lines represent median values and error bars represent a 95% confidence interval. (C) Midgut collagen IV damage 96 hours after 1, 2, 3, or no blood meal using pools of 3 midguts with 11–13 pools per treatment (Unfed n = 13, BM1 n = 11, BM2 n = 12, BM3 n = 13). Pools were tested for CHP binding and evaluated using a one-way ANOVA with a Tukey's multiple comparisons post-test. Center lines represent means and bars represent SEM. On graphs, n.s. = not significant, (*) p<0.05, (**) p<0.01, (***) p<0.001, and (****) p<0.0001. Exact p values can be found in text.

## Discussion

As an important arbovirus vector, *Ae. aegypti* feeds frequently and often takes partial or mixed blood meals which has cumulative effects on mosquito physiology, vector competence, and virus transmission dynamics in the field. Through our experiments we sought to better mimic natural *Ae. aegypti* feeding behavior and study the impact of blood meal size on basal lamina damage and virus dissemination. We further investigated the link between basal lamina damage and the midgut escape barrier by quantifying DENV-2 dissemination in mosquitoes given a minimal blood meal to mimic the unfed state. Expanding on previous experiments we investigated basal lamina collagen IV damage and repair in mosquitoes following varying numbers of blood meals.

Here we show that midgut basal lamina collagen IV damage is dependent on blood meal size and midgut expansion. Mosquitoes that were given minimal blood meals had a level of damage similar to that detected in unfed mosquitoes (Fig 3B), partial blood meals led to an intermediate level of damage (Fig 1B), and full blood meals produced the highest level of damage (Fig 1B). Although basal lamina damage as measured by CHP assay scales with degree of midgut expansion, we still detected collagen IV damage in unfed midguts (Figs 1B, 3B, 4B, and 4C). This may be the result of nonspecific binding by CHP or may indicate that even naïve mosquitoes have an inherent baseline level of basal lamina collagen IV damage or that persistent extracellular matrix renewal results in detectable collagen IV damage [19]. Although the cause of this damage detected in unfed midguts is unknown, midgut damage as measured by CHP assay matched that observed through TEM imaging with unfed midguts having dense basal lamina layers, partially fed mosquito midguts having more low density areas and disorder, and fully fed midguts having the most disorder (Fig 1C and 1D). Others have noted virus particles in low density areas of the basal lamina and observed disorder following blood feeding, indicating a possible role for this disorder in viral escape [3,5]. One group also saw a decrease in basal lamina thickness following blood feeding that they attributed to stretching during midgut expansion [3]. Although muscle stretching was clear in SEM images (S1 Fig), in contrast to this previous work, we found that increased basal lamina disorder and low density areas expanded the width of the basal lamina layer following blood feeding (Fig 1C and 1D) [3]. While disorder may be further elevated around visceral musculature as suggested by others, we found similar differences between treatment groups when comparing basal lamina width measurements from a variety of sections (Fig 1D) as well as when exclusively measuring the basal lamina width in areas without visceral muscles (S2 Fig) [3]. Further, we believe that the inclusion of basal lamina thickness measurements from areas both with and without visceral muscles better captures the physiological changes that occur across the midgut following expansion and areas near muscles may even represent areas of increased virus midgut escape [3,14].

After finding evidence of increasing basal lamina damage with blood meal size and in light of previous data showing earlier virus dissemination when mosquitoes were given a second full blood meal, we examined the impact of secondary partial blood meals on virus dissemination and midgut escape [2]. When mosquitoes were given a full primary infectious DENV-2 feed, all groups became infected at the same rate regardless of secondary blood meal presence or size (Fig 2C) as shown previously [2]. Further, mosquitoes that received a full or partial blood meal three days after an infectious feed had higher rates of virus dissemination when compared to mosquitoes that did not receive a second blood meal (Fig 2D). This shows the robust effect of even partial secondary blood meals on virus dissemination. Mosquito leg dissemination assays, as used in this study, were previously shown to be more accurate at predicting virus transmission to a vertebrate host than traditional forced salivation assays and earlier

virus midgut escape leads to earlier transmission [20]. As *Ae. aegypti* feed frequently and partial or mixed blood meals are common, vector competence studies need to account for this feeding behavior and determine the influence it can have on the timing of virus dissemination and subsequent transmission. By relying on data from mosquitoes fed a single blood meal, we may be overestimating EIP and our data suggests that even small secondary blood meals may enhance dissemination and shorten time to transmission.

Despite many studies, it is unclear how viruses escape the mosquito midgut or if damage or stress from blood feeding is essential for this escape to occur. We have noted that while blood fed mosquitoes have transient midgut damage, this damage decreases quickly and is on par with that observed in unfed mosquitoes by 48–96 hpbm via CHP assay (Fig 4C) [2]. Additionally, damage does not appear to persist between meals as mosquitoes given an initial noninfectious blood meal followed by a secondary infectious feed had similar dissemination dynamics as mosquitoes given a single infectious feed [2]. As dissemination from the midgut occurs after midgut damage has returned to a level found in unfed mosquitoes, this suggest that the midgut is not entirely impermeable even at a baseline or unfed state despite having pore sizes smaller than a virus particle [2,12,13]. Contrary to this, previous work showed a lack of midgut invasion from the hemolymph 3 or 4 dpi when chikungunya or Zika virus was intrathoracically injected into the hemolymph of unfed mosquitoes [5]. In this case, blood feeding following injection of virus into the hemplymph made midgut invasion permissible, suggesting that blood feeding damage or stress is critical for altering midgut permeability [2,5]. To further clarify this relationship between blood feeding, basal lamina damage, and dissemination, we gave mosquitoes minimal blood meals to mimic a naïve, unfed state. While our limits of detection via CHP assay are somewhat unknown, mosquitoes given minimal blood meals had levels of midgut expansion and basal lamina damage 0 hpbm equivalent to those in mosquitoes that were unfed (Fig 3A and 3B). When given a minimal infectious feed of DENV-2 diluted 1:15 with defibrinated sheep's blood, mosquitoes still became infected and virus dissemination occurred even in the absence of detectable midgut basal damage 0 hpbm (Fig 3C). While other processes may account for some virus dissemination, this dissemination in the absence of increased basal lamina damage and the consistent degree of damage detected in unfed and repaired mosquito midguts via CHP assay, lend support to the idea that the midgut basal lamina is not entirely impermeable following adult emergence and may have a baseline level of damage or that renewal and turnover can also allow for dissemination [14,19].

Although blood feeding induces a transient increase in collagen IV triple helix unfolding, this damage quickly returns to baseline levels through repair or replacement processes that are poorly understood. While, during *Drosophila* development and embryogenesis, basal lamina proteins are thought to be secreted by epithelial cells, fat body cells, or hemocytes depending on the underlying tissue and developmental stage, the sources of these proteins are poorly characterized [21]. One matricellular protein known as Secreted Protein Acidic and Rich in Cysteine (SPARC), is expressed by hemocytes and fat body cells during *Drosophila* development, and binds to hemocyte-derived collagen IV, facilitating collagen IV integration into the developing basal lamina [22–25]. While SPARC is critical for collagen IV integration during basal lamina development and larval survival, in adult mice SPARC expression is associated with tissues that undergo a high rate of turnover and remodeling (such as the gut) and may play a role in basal lamina turnover [26,27].

Although SPARC expression in mosquitoes has been poorly characterized, there is some evidence to suggest that damaged collagen IV in the basal lamina is replaced following blood feeding, possibly with the help of SPARC and remodeling enzymes. Previous work demonstrated that collagen IV levels in the midgut basal lamina decreased following blood feeding while collagenase activity increased, yet by ~48 hpbm collagen IV levels recover, mirroring our

timeline of collagen IV damage as assessed through CHP assay and suggesting a removal and replacement process for damaged collagen IV [2,5]. This work also found that overexpression of *Ae. aegypti* tissue inhibitor of metalloproteinases (AeTIMP) led to diminished matrix metalloproteinase (MMP) activity and increased CHIKV escape from the midgut suggesting that MMPs such as MMP1 that possess collagenase activity may be important for facilitating midgut basal lamina repair [5].

Additional findings in mosquitoes and *Drosophila* support the idea that basal lamina remodeling and restoration depends on a combination of enzymatic activity, hemocytic processes, intestinal stem cell dynamics, and the secretion and integration of new proteins. Hemocyte numbers, size, and level of immune activation increase following blood feeding in mosquitoes and, in *Drosophila*, hemocytes migrate to the gut and increase intestinal stem cell proliferation following injury [28–31]. While more work is needed to determine how the basal lamina is repaired or renewed in mosquitoes, it is possible that following blood feeding, damaged collagen IV is removed from the basal lamina through the action of various enzymes including MMPs while hemocytes proliferate and are recruited to the midgut [19,32]. At the same time MMPs and hemocytes impact intestinal stem cells proliferation and gut integrity [30,33]. New collagen IV bound to SPARC may be synthesized by hemocytes, midgut epithelial cells, or the fat body and used to replace the removed damaged collagen IV, returning the basal lamina back to baseline [19,32]. A similar process may occur but on a smaller scale during baseline basal lamina turnover.

To further study midgut repair and the influence of frequent *Ae. aegypti* blood feeding, we compared midgut basal lamina damage 24 hpbm and 96 hpbm in mosquitoes given either no blood meal or 1, 2, or 3 blood meals with each feeding separated by four days. Through this we found that midgut basal lamina damage 24 hpbm increases slightly in a non-significant manner with the number of blood meals (Fig 4B) although damage appeared to return to a baseline level of damage similar to that in unfed midguts by 96 hpbm (Fig 4C). As all mosquitoes were the same age in this study and were fed on a staggered schedule to allow for concurrent midgut harvesting, this slight increase in damage with blood meal number is independent of age and may instead indicate that we are missing persistent damage following blood feeding due to a lack of CHP assay sensitivity. Although collagen IV damage seems to be repaired, it is also possible that there is damage to other midgut basal lamina proteins not measured by our assay that persist, making the gut less elastic and more susceptible to damage with additional expansions. Alternatively this non-significant increase in damage may be an artifact of sampling and it is unclear if this has any biological significance or impact on viral dynamics within the mosquito.

Together, our attempts to recapitulate wild *Ae. aegypti* feeding behavior in this study have yielded several interesting findings. We observed a strong connection between blood meal size and basal lamina damage while demonstrating that partial blood meals are sufficient to have a profound impact on virus dissemination. We have also shown that the number of blood meals taken by a mosquito influences basal lamina damage levels shortly after blood feeding and perhaps impacts EIP. Spatiotemporal aspects of basal lamina damage particularly as relating to unfed midgut permeability and virus dissemination following varying numbers of blood meals deserve more attention; however, our data suggest that the midgut basal lamina has a baseline level of damage even in naïve mosquitoes. This baseline level of damage still allowed for virus dissemination, indicating that the mosquito midgut is not impermeable even in unfed or minimally fed mosquitoes. These data highlight the connection between mosquito feeding behavior and virus transmission dynamics and demonstrate the importance of taking mosquito behavior in the wild into consideration when conducting laboratory studies. By failing to incorporate wild mosquito feeding behavior in laboratory study design we may be overestimating EIP

or overlooking ways that viruses persist when vector populations appear less competent via traditional vector competence assays. Mosquito feeding behavior and the effect such behavior can have on EIP need to be taken into consideration when designing control programs, future laboratory experiments, or when modeling outbreaks of arboviruses.

## Supporting information

**S1 Fig. Blood feeding leads to muscle stretching and may compromise gut integrity.** Overall midgut shape as well as stretching of muscles (arrows) surrounding midguts from mosquitoes given no blood meal (Unfed), a partial blood meal (Partial), and a full blood meal (Full). (TIF)

**S2 Fig. Basal lamina thickness is increased with blood feeding even in areas where muscles are not present.** Basal lamina thickness 24 hpbm measured in areas that were not close to surrounding muscles. Measurements were from 2 mosquitoes per treatment with 3 TEM images per mosquito and 5 measurements per image that were averaged for each image to give 6 measurements per treatment (n = 6). Differences were assessed using a one-way ANOVA with a Tukey's multiple comparisons post-test. On graph, center lines represent means and bars represent SEM. Exact p values for treatment comparisons are as follows: Unfed vs Partial p = 0.0776, Unfed vs Full p <0.0001, and Partial vs Full p = 0.0044. (TIF)

**S1 Text. Supporting information.**
(DOCX)

## Acknowledgments

We would like to thank Dr. Xinran Liu and the Yale Center for Cellular and Molecular Imaging Electron Microscopy Facility for their help preparing and imaging midguts.

## Author Contributions

**Conceptualization:** Rebecca M. Johnson, Zannatul Ferdous, Philip M. Armstrong, Doug E. Brackney.

**Data curation:** Rebecca M. Johnson, Duncan W. Cozens, Zannatul Ferdous.

**Formal analysis:** Rebecca M. Johnson, Duncan W. Cozens, Zannatul Ferdous.

**Funding acquisition:** Philip M. Armstrong, Doug E. Brackney.

**Investigation:** Rebecca M. Johnson, Duncan W. Cozens, Zannatul Ferdous.

**Methodology:** Rebecca M. Johnson, Duncan W. Cozens, Zannatul Ferdous.

**Supervision:** Philip M. Armstrong, Doug E. Brackney.

**Writing – original draft:** Rebecca M. Johnson.

**Writing – review & editing:** Rebecca M. Johnson, Philip M. Armstrong, Doug E. Brackney.

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
