## [Decision Letter · Decision Letter 0]

30 Jul 2023

Dear Dr Brackney,

Thank you very much for submitting your manuscript "Increased blood meal size and feeding frequency compromise mosquito midgut integrity and enhance arbovirus dissemination" for consideration at PLOS Neglected Tropical Diseases. As with all papers reviewed by the journal, your manuscript was reviewed by members of the editorial board and by several independent reviewers. In light of the reviews (below this email), we would like to invite the resubmission of a significantly-revised version that takes into account the reviewers' comments. 

The three reviewers have provided excellent recommendations for revising the manuscript. Namely, key information about viral titer is missing from the methods and results - hopefully this will be straightforward to clarify. The reviewers requested minor improvements in language and grammar, as well as making some helpful clarifications to the figures. The authors are encouraged to modify the discussion as suggested by the reviewers, perhaps clarifying the "big picture" in terms of turnover of gut epithelia and how that relates to the findings here and in Armstrong et al.

My sincere apologies for the delay in processing this manuscript - one peer reviewer "went dark" at the last moment and we were forced to recruit more. Thank you for your patience and understanding - JVC

We cannot make any decision about publication until we have seen the revised manuscript and your response to the reviewers' comments. Your revised manuscript is also likely to be sent to reviewers for further evaluation.

Sincerely,

Jeremy V. Camp, Ph.D.

Academic Editor

Abdallah Samy

Section Editor

The three reviewers have provided excellent recommendations for revising the manuscript. Namely, key information about viral titer is missing from the methods and results - hopefully this will be straightforward to clarify. The reviewers requested minor improvements in language and grammar, as well as making some helpful clarifications to the figures. The authors are encouraged to modify the discussion as suggested by the reviewers, perhaps clarifying the "big picture" in terms of turnover of gut epithelia and how that relates to the findings here and in Armstrong et al.

My sincere apologies for the delay in processing this manuscript - one peer reviewer "went dark" at the last moment and we were forced to recruit more. Thank you for your patience and understanding - JVC

Reviewer's Responses to Questions

**Key Review Criteria Required for Acceptance?**

**Methods**

-Are the objectives of the study clearly articulated with a clear testable hypothesis stated?

-Is the study design appropriate to address the stated objectives?

-Is the population clearly described and appropriate for the hypothesis being tested?

-Is the sample size sufficient to ensure adequate power to address the hypothesis being tested?

-Were correct statistical analysis used to support conclusions?

-Are there concerns about ethical or regulatory requirements being met?

Reviewer #1: Additional details must be added to the methods regarding the high titer infections prior to publication.

Reviewer #2: The authors do not provide any reference for the method used to estimate the collagen hybridization assay they used. Also, they do not mention if the assay has been validated for insect tissues in general or for mosquitoes specifically. Also, it appears to me that this probe does not seem to be specific for collagen IV, but the authors assume that what they are measuring is only type IV. In the lack of fluorescence microscopy showing label is only at the BM, this statement may be just an hypothesis (although very likely). 

They do not give information on the amount of virus (PFU, preferably) used in the infection assays, what difficult comparison with data from the literature. 

Also, in line 315 , they use a “high” concentration of DENV-2, but the reader receive no information on what they mean by high (or low) … 

The viral titers are evaluated only by qPCR, and not as infective particles, what is a weak point of the paper. 

Partial feeding (and minimal feeding as well) is observed qualitatively. This is ok for a routine protocol, but it would be useful for replication of their work by others if they quantify this partial as a range of blood volume (measured as weight, or hemoglobin or protein content compared to fully fed condition).

Replication description is not clear in several parts of the paper. The major point here is to make clear the number of mosquitoes measured or the number of pools used. This is the true n. Fig 1 D, as an example, say thy collected 10 images per treatment, five measurements per image, but do not mention the number of insect replicates. In panel 1D, each point is a different pool of three mosquitoes?

Reviewer #3: (No Response)

**Results**

-Does the analysis presented match the analysis plan?

-Are the results clearly and completely presented?

-Are the figures (Tables, Images) of sufficient quality for clarity?

Reviewer #1: Some small changes are needed in the figures to increase clarity.

Reviewer #2: • They use qPCR just to say if the insect is infected or not. Of course, their raw data should include the amount of virus, what is a relevant information. This information could be made available, at least as a supplementary information. 

• In Figure S2, while the bar size seems correct, the amplification apparently is about 10 X what should be. The bottom left photo certainly is not a 613 X magnification. 

• What is the red arrow in Figure S2? Structures present in the figure should be labelled.

• In several parts of the text the expression “infection levels “(ex: line 273) is used to refer to % of infection (prevalence). However, infection level is used more frequently to refer to amount of virus, so I suggest using this term for clarity. 

• Figure 2 legend says Midgut DENV-2 infection, but in methods only whole-body assays are mentioned

Reviewer #3: (No Response)

**Conclusions**

-Are the conclusions supported by the data presented?

-Are the limitations of analysis clearly described?

-Do the authors discuss how these data can be helpful to advance our understanding of the topic under study?

-Is public health relevance addressed?

Reviewer #1: Yes

Reviewer #2: • I have two observations that I left to the authors to use or not, but I would like to share my thoughts on the subject with the authors. They look to CHP binding as an index of damage, and only in a single instance mention that this binding may be related to BM turnover. However, gut epithelia have probably the highest cellular turnover in most organisms. So, what they call damage, may also be described as use-related stress, followed by renewal. This would call the need not only for extracellular proteins (not only collagen) but also for cell death coupled to stem cell differentiation and division. 

• Additionally, if we are talking about compromised barrier function of the gut upon distension, this may increase exposition of the body cavity to microbial elicitors, and trigger immune response that may lead to self-inflicted damage beyond the purely mechanical damage they assume in their model. 

• The discussion is too extensive and frequently recapitulates the results section. 

• Some parts are too speculative, such as the one and half pages discussing SPARC function, for which there is no data in the paper.

Reviewer #3: (No Response)

**Editorial and Data Presentation Modifications?**

Reviewer #1: Major Revision

Reviewer #2: In fig 1 B and D panels, the center lines should be placed in front of the data points in the D full data it is not possible to identify the position of the line because is it hidden behind the data points. There is no error bar visible in the figure, although it is mentioned in the legend.

Reviewer #3: (No Response)

**Summary and General Comments**

Reviewer #1: Comments to Authors

The manuscript by Johnson et al examine the effects of blood meal size, frequency, and repair of the midgut basal lamina in the context of arbovirus infection. The results are innovative and novel as they continue to explore the previously described effects of multiple blood meals on virus dissemination, providing new insight into the likely mechanisms of virus transmission in field settings where Aedes mosquitoes feed frequently after taking an infectious blood meal. The results are straight-forward and well-executed, such that the contributions will add a valuable piece to our understanding of mosquito vector competence. I have only a few comments which should improve clarity of the manuscript prior its publication. 

Major Comments

-For the data in Figure 1D and Figure S1, it isn’t clear what data are displayed. If I understand correctly, there were 10 images with 5 measurements per image. It appears as though the authors display this as 50 points. I would argue this is artificial and unjustly increases the statistical power for these observations. I would argue that the repeat measures per image should be averaged, with data displayed for each of the 10 TEM images. This will likely change the statistical outcomes, but would argue this is a more accurate depiction of the limited data points. 

-The authors describe a “minimal blood meal” with concentrated virus, yet experimental details as to how this was performed are lacking in the methods. How do these “high titers” compare to regular infections? Infection titers should be included. This should be corrected for rigor and reproducibility.

-The authors present all of their infection data only as prevalence, without mention of viral titers. How does the infection intensity compare to prevalence? Does a second blood meal (full or partial) influence infection? For transparency, all infection data should be included in the supplement as supplementary figures. 

Minor Comments

-For Figure 1B, please display the number of pools examined in either the figure or figure legend.

-In Figure 2C and 2D, it would be helpful to describe the tissues being evaluated. Based on 2A, it seems that this is the carcass (body) and legs being examined, but this is not explicitly stated in the figure or figure legend.

-Figure 4A: The graphic shows harvesting body and legs similar that in Figure 2. However, I would argue that this is misleading since only midguts are being examined. Please correct. 

-For Figure 4B and 4C, I suggest respectively adding 24h and 96h to the figures to denote what the data represent without having to solely rely on the figure legend. 

-Figure S2: It is unclear what the arrows are pointing to. There is no mention in the figure legend.

Specific Comments

-Line 86-87: What is meant by “often obscured by common mosquito sorting practices”? Please elaborate and cite any relevant references.

-Line 106-108: Please revise sentence “In singly fed mosquitoes” for clarity. Try to refrain from use of multiple adjectives in the same sentence (such as decreasing, beginning, lowering, etc.) to improve clarity.

-Line 401: There is mention of “in contrast to previous studies”, yet there are no citations as to which the authors might be directly referencing.

-Lines 434-438: I found these sentences confusing. Please revise for clarity. 

-Lines 478-483: Long run-on sentence. Revise for clarity.

Reviewer #2: This paper is a follow-up for the previous report Armstrong et al. In the first communication they made an important contribution that was showing that a second, even non infective blood meal within a short interval enhances viral transmission, proposing that this was due to microperforations in the basal lamina. Here they add a relevant complement to that initial report, while showing that even partial blood meals were capable of that action. Overall, this is still a relevant contribution, pointing to novel variables that should be taken into account to evaluate vector competence under realistic field conditions.

Reviewer #3: In this manuscript, the authors have examined the effects of partial blood meals on Aedes aegypti midgut basal lamina integrity and on dengue virus dissemination. The authors have tested whether partial blood meals, where the mosquito has not fed to repletion, will enhance virus dissemination similar to what they previously published regarding full blood meals. They claim that this more accurately reflects the natural situation where mosquitoes often are not able to take full blood meals due to host defensive behavior. Using a collagen binding assay and TEM to assess damage to basal lamina, the findings indicate that larger blood meals have more damaging effects than partial blood meals, but that partial meals still cause an intermediate level of damage. These effects correlate to levels of DENV dissemination. Interestingly though, increased damage is not required for midgut escape, since some proportion of minimally fed mosquitoes that do not have more basal lamina damage than unfed mosquitoes still develop disseminated infection. Also, even in fully fed mosquitoes the damage appears to have been resolved by the time that DENV escapes the midgut. Taken together these results provide some additional insights into the mechanisms of arbovirus midgut escape. I have the following specific comments:

1. The title is overly broad. The authors have only examined one mosquito species and one arbovirus. Suggest changing to “Increased blood meal size and feeding frequency compromise Aedes aegypti midgut integrity and enhance dengue virus dissemination”

2. A weakness of the study is that the dose of virus fed to the mosquitoes is unknown. It is not clear why the virus stocks used to infect mosquitoes were not quantified by RT-qPCR. This makes it more difficult produce consistent results, both within the laboratory and in other laboratories who may want to repeat these experiments or use this approach.

3. Line 345, BM1 should be BM2

4. Line 393, the authors refer to “persistent extracellular matrix renewal as has been seen in humans”. It is my impression that all extracellular matrix is continuously being turned over, as it is constantly degraded and resynthesized. The authors should look into this in more depth. For example, see review article PMC3225943.

5. The order of the two supplemental figures appears to be reversed compared to how they are described in the legends and text.

6. Line 541, National Institutes of Health

PLOS authors have the option to publish the peer review history of their article (what does this mean?). If published, this will include your full peer review and any attached files.

Reviewer #1: No

Reviewer #2: No

Reviewer #3: No
---

## [Decision Letter · Decision Letter 1]

5 Oct 2023

Dear Dr Brackney,

We are pleased to inform you that your manuscript 'Increased blood meal size and feeding frequency compromise Aedes aegypti midgut integrity and enhance dengue virus dissemination' has been provisionally accepted for publication in PLOS Neglected Tropical Diseases.

Best regards,

Jeremy V. Camp, Ph.D.

Academic Editor

Abdallah Samy

Section Editor

Reviewer's Responses to Questions

**Key Review Criteria Required for Acceptance?**

**Methods**

-Are the objectives of the study clearly articulated with a clear testable hypothesis stated?

-Is the study design appropriate to address the stated objectives?

-Is the population clearly described and appropriate for the hypothesis being tested?

-Is the sample size sufficient to ensure adequate power to address the hypothesis being tested?

-Were correct statistical analysis used to support conclusions?

-Are there concerns about ethical or regulatory requirements being met?

Reviewer #1: (No Response)

Reviewer #2: This is the second review. Critical points were fixed. Methods are appropriately described now.

Reviewer #3: (No Response)

**Results**

-Does the analysis presented match the analysis plan?

-Are the results clearly and completely presented?

-Are the figures (Tables, Images) of sufficient quality for clarity?

Reviewer #1: (No Response)

Reviewer #2: This is the second review. Essential points were well addressed

Reviewer #3: (No Response)

**Conclusions**

-Are the conclusions supported by the data presented?

-Are the limitations of analysis clearly described?

-Do the authors discuss how these data can be helpful to advance our understanding of the topic under study?

-Is public health relevance addressed?

Reviewer #1: (No Response)

Reviewer #2: This is the second review. Essential points were well addressed

Reviewer #3: (No Response)

**Editorial and Data Presentation Modifications?**

Reviewer #1: (No Response)

Reviewer #2: This is the second review. Essential points were well addressed

Reviewer #3: (No Response)

**Summary and General Comments**

Reviewer #1: My previous comments have been addressed. I recommend that the article is accepted.

Reviewer #2: This is the second review. Essential points were well addressed.

The paper represents a relevant follow up report on a previous work frm the authors.

Reviewer #3: I am satisfied with the changes made to the manuscript, which has been improved significantly.

PLOS authors have the option to publish the peer review history of their article (what does this mean?). If published, this will include your full peer review and any attached files.

Reviewer #1: No

Reviewer #2: No

Reviewer #3: No

---

## [Editor Report · Acceptance letter]

18 Oct 2023

Dear Dr Brackney,

We are delighted to inform you that your manuscript, "Increased blood meal size and feeding frequency compromise Aedes aegypti midgut integrity and enhance dengue virus dissemination," has been formally accepted for publication in PLOS Neglected Tropical Diseases.

Best regards,

Shaden Kamhawi

co-Editor-in-Chief

Paul Brindley

co-Editor-in-Chief
